# Anthocyanins Potentially Contribute to Defense against Alzheimer’s Disease

**DOI:** 10.3390/molecules24234255

**Published:** 2019-11-22

**Authors:** Mohammad Afzal, Amina Redha, Redha AlHasan

**Affiliations:** Biological Sciences Department, Faculty of Science, Kuwait University, Safat-13060, Kuwait; aminaredha@gmail.com (A.R.); redha49@gmail.com (R.A.)

**Keywords:** anthocyanins, antioxidants, transition metals, reactive oxygen species

## Abstract

Anthocyanins (ANTs) are plant pigments that belong to a flavanol class of polyphenols and have diverse pharmacological properties. These compounds are primarily found in fruits and vegetables, with an average daily intake of 180 mgd^−1^ of these compounds in the developed world. ANTs are potent antioxidants that might regulate the free radical-mediated generation of amyloid peptides (Abeta-amyloids) in the brain, which causes Alzheimer’s disease (AD). This study presents a literature review of ANTs from different berries and their potential therapeutic value, with particular emphasis on neurodegenerative AD, which owing to oxidative stress. This review also highlights reactive oxygen species (ROS) generation through energy metabolism, nitrogen reactive species, the role of transition metals in generating ROS, and the radical-quenching mechanisms of natural antioxidants, including ANTs. The current status of the bioavailability, solubility, and structure activity relationship of ANTs is discussed herein.

## 1. Introduction

By definition, berries are small fleshy fruits produced from a single ovary. Berries from eight plant species are considered the healthiest fruit for consumption, namely strawberries, acai berries, grapes, blueberries, goji berries, bilberries, raspberries, and cranberries. Most berries are rich in non-nutritive/nutritive, water-soluble, bioactive flavonoid polyphenols. These include anthocyanidin (sugar-free anthocyanin (ANT)), ANT-3-glycosides, 5-glycosides, diglucosides, acylated ANTs, and enzymatically formed flavanol-ANTs adducts (vitisins). Pyrano-anthocyanins are formed by adding pyruvic acid and acetaldehyde to anthocyanin molecules. In ANTs, glycosylation might occur at positions 3′, 4′ and 5′, as shown in Figure 1 [1]. These molecules have a common flavylium cation structure that generates color. ANTs are present in plant cell vacuoles, and their color is cell-sap pH-dependent [2,3]. *Conocarpus lancifolius* is a wild plant resistant to biotic and abiotic stress in the desert of Kuwait, and its berries are very rich in various anthocyanins (unpublished work of the authors). There are approximately 17 naturally occurring ANTs, six of which are very common. These include ANT glycosides/diglucosides of cyanidin, delphinidin, malvidin, pelargonidin, peonidin, and petunidin (Figure 1). These ANTs also contain the six most common sugar residues, namely glucose, rhamnose, galactose, arabinose, xylose, and rutinose [2] (Figure 1). To-date, more than 6000 flavonoid polyphenols have been reported [4]. Epidemiological and preclinical studies have shown that the polyphenols of red wine contribute in modulating neurodegenerative diseases, such as Alzheimer’s disease (AD) [5]. One such polyphenol is quercetin-3-*O*-glucoside (QG), which is a component of many vegetables and fruits, including apples, grapes, black tea, green tea, broccoli, and red onions. It is a potent brain-targeting bioflavonoid that is beneficial in limiting AD. A QG metabolite, quercetin-3-*O*-glucuronide, accumulates in the brain, significantly reducing Abeta 1–40 and Abeta 1–42 toxicity. At present, many health supplements containing quercetin are available over the counter. Beta-secretase is a rate limiting enzyme that forms a number of Abeta peptides from the amyloid precursor protein to yield Abeta 40, Abeta41, and Abeta 25–35. These Abeta peptides are involved in amyloid-mediated neurodysfunction [6].

To date, more than 650 ANTs have been identified from fruits and vegetables [7]. ANT micronutrients have many beneficial health effects, including cancer-preventive, immuno-protective, and neuroprotective effects; moreover, they prevent cognitive function impairment, alleviate inflamed and arduous joints, and attenuate rotundity, alopecia, dementia, cardiovascular diseases, and obesity [8,9,10,11,12,13]. Blackberries (*Rubus sp*.), mulberries (*Morus alba* L.), and elderberries have antimutagenic, antimicrobial, anti-inflammatory, antidiabetic, and anti-neurodegenerative activities [14,15]. These activities are associated with their potent antioxidant actions, which help control oxidative stress (OS) due to aging. These ANTs inhibit metallo-proteinases 2 and 9, which are associated with the metastasis of cancerous cells, and cisplatin-induced renal pathophysiology [16,17,18,19]. The in vitro effect of ANTs on metabolic syndrome-related enzymes and anxiety has also been reported [20,21].

Berries are also rich in bioflavonoids and ellagitannins, which are naturally found in fruits and vegetables, particularly in the white core of citrus fruits. Bioflavonoids and their glycosides are structurally related to anthocyanidins and ANTs, affecting a wide range of biological systems. These molecules are the main antioxidants in plants used to treat allergies, viral infections, arthritis, and anti-inflammatory conditions, including hemorrhoids.

Quercetin (Figure 2) is a polyphenolic bioflavonoid that is highly concentrated in red wine, grapes, broccoli, citrus, and red and yellow onions. In human breast cancer cell lines, quercetin is an agonist of the G protein-coupled α,β-estrogen receptors, which is activated by 17β-estradiol, in addition to regulating different genes and modulating learning and memory [22,23]. In rats, quercetin does not undergo phase I metabolism; however, in phase II metabolism, it produces many conjugated metabolites, with five of its hydroxyl groups glucuronidated by UDP-glucuronosyl transferase. Quercetin-3-*O*-glucosides (Q-3-G) exhibit estrogenic-like behavior with non-specific proteins kinase inhibitor activities, and they promote estrogenic receptors [22,24]. Quercetin, its 3-*O*-glucoside and its metabolite quercetin-3-*O*-glucuronide are brain-targeting molecules that may be effective against AD [5]. Similarly, pelargonidin (Pel) ANT (Figure 1) is an estrogen receptor agonist that has fewer side effects than those of estrogen [25]. Pel-ANT is a neuroprotective agent that protects against amyloid Abeta25-35 in rat models of AD. Pel-ANT diminishes malon-dialdehyde, a lipid peroxidation index. Thus, Pel-ANT decreases OS, increases the activities of catalase and AChE, decreases the toxicity of Abeta25-35, and regulates the cognitive functions of the brain through the estrogen receptors [25,26]. To-date, no effective therapeutic agent has been developed that can reverse neuron damage caused by AD. Considering the global prevalence of AD, it is vital to develop new drugs to mitigate the ever-increasing pathologies of AD.

ANTs have been used in traditional medicine since antiquity. Yet, the physiology of ANTs remains poorly defined, and their mechanisms of action (i.e., absorption, uptake, and bioavailability) remain poorly understood. However, many studies have strongly supported their health benefits against various diseases, including cancer, cognitive decline, and cardiovascular diseases. This review focuses on how ANTs can be used to alleviate cognitive issues in AD.

## 2. Alzheimer’s Disease

AD was first described in early 1907 by a German psychiatrist, Alois Alzheimer. It is a progressive neurodegenerative disease that worsens with advancing and is responsible for 60–70% of dementia cases [27,28]. By 2050, it is predicted that 1 in 86 people will have AD [29]. In general, the global population is aging, and the projected increase in AD is linked to the increasing age of people. AD is not a simple psychosis dementia due to the normal aging process; instead, it has conclusive neuropathology characterized by the loss of neurons and synapsis in the cerebral cortex. The degeneration of the temporal and parietal lobes with an atrophy is the consequence of AD [30]. AD is accompanied by mitochondrial dysfunction, release of lactate dehydrogenase, increased ROS, decreased superoxide dismutase activity, and increased intracellular calcium, as well as modifications of neuronal morphology [31]. Dementia might also be caused by other abnormalities, such as vascular pathologies, Lewy bodies, frontotemporal lobar degeneration, or Parkinson’s disease (PD) [27,32]. Early symptoms include disorientation, language problems, loss of motivation, behavioral issues, mood swings, agnosia, apraxia, aphasia, and lack of self-care. Many patients with AD withdraw from their families and the community, preferring solitude, which leads to bodily dysfunctions and death. The typical life expectancy after AD diagnosis is 3–7 years [33,34].

The cause of AD is not well understood. Recently, Hansson et al. (2018) have reported accurate measurement of tau/Aβ ratios with PET scan [35]. Histological brain scans have also been used to diagnose AD. In general, AD is a disease caused by the misfolding of Aβ and neurofibrillary tangles of hyperphosphorylated tau-proteins, generating a neurotic senile plaque in the brain [36,37]. The cause of AD is not well understood. Recently, Hansson et al. (2018) have reported accurate measurement of tau/Aβ ratios with PET scan [35]. Histological brain scans have also been used to diagnose AD. In general, AD is a disease caused by the misfolding of Aβ and neurofibrillary tangles of hyperphosphorylated tau-proteins, generating a neurotic senile plaque in the brain [36,37]. The transmembrane protein amyloid precursor protein (APP) is cleaved by beta- and gamma-secretase activities to generate Abeta, with 39–43 amino acids [38]. The amyloidogenic processing of the APP towards the more toxic Abeta42 is increased in dominant forms of AD, while no changes in total APP levels occur. APP is essential for neuronal growth, survival, and damage repair [39]. Apart from genetic causes, cholinergic, Abeta 1–42 peptide, or a closely related protein (such as abnormally aggregated and hyperphosphorylated pathogenic tau-protein) triggers abnormalities in the brain, resulting in depression, hypertension, or head trauma with neuronal death [27,28,40,41,42]. The beta-amyloid protein precursor (AβPP) is located at chromosome 21, and the progression of the disease causes plaque and neurofibrillary complications in the brain. Ten percent of patients suffer from AD due to mutations in the Abeta protein precursors (APP), presenilin-1 and presenilin-2, which give rise to Abeta peptides [38].

Although many pathways have been suggested to underlay the development of AD, no precise mechanism has been discovered. Astrocytes play important roles in the inflammatory/immune reaction of the central nervous system (CNS). Gonzalez-Reyes et al. [43] reported changes in the functioning of astrocytes during AD, with both astrocytes and Abeta1-42 implicated in the disruption of gliotransmission, uptake of neurotransmitters, and calcium signaling in astrocytes. Astrocytes are also involved in the expression of apolipoprotein E and the degradation and removal of Abeta1-42, indicating that astrogliosis occurs during the progression of AD [43]. Abeta is produced through oxidative damage of astrocytes, affecting intracellular calcium levels, NADPH oxidase (NOX), NF-kappaB signaling, exciter glutamate uptake, and mitochondrial function [43].

## 3. Alzheimer’s Disease and Oxidative Stress

Aging is one the biggest risk factors of AD, following OS. A super oxide anion (O_2_^−**.**^) radical is generated in the mitochondrial electron transport chain that forms H_2_O_2_, which then generates a cytosolic hydroxyl radical (HO**^.^**). In the outer membrane of the mitochondria, HO**^.^** initiates monoamine oxidase to catalyze the oxidation of biogenic amines, leading to the further formation and supply of H_2_O_2_, generating reactive oxygen species (ROS). These oxygen species trigger a chain of radical events that disrupt neuronal membranes; various biomolecules—such as DNA, RNA, and amyloid β-peptides; and lipid peroxidation [44]. The oxidation of cytoplasmic DNA is prevalent, rather than that of nuclear DNA. Decreased plasma levels of antioxidants (such as albumin, bilirubin, uric acid, lycopene, and vitamins C and E) have been implicated as neurodegenerative disease pathologies in AD [45,46,47]. Moreover, the activity of antioxidant enzymes (such as superoxide dismutase, catalase, glutathione peroxidase, and heme-oxygenase) decreases in AD patients [4,47,48]. The brain is a major organ that consumes 20% of the body’s oxygen. It can generate super oxide anion (O_2_^−**.**^) radicals and ROS, thereby disturbing cellular redox balance. This phenomenon triggers an imbalance in tissue levels of ROS and antioxidants. In turn, this process triggers detrimental OS in the brain, causing mitochondrial dysfunction and Aβ to aggregate, leading to the pathogenesis of AD. Thus, the main cause of AD is an imbalance between the production of ROS and their quenching by antioxidants and related enzymes [49,50]. The effect of oxidative burst in the brain is shown in Figure 3.

Evidence suggests that the age-dependent accumulation and deregulation of the redox metals Fe (III) and Cu (II), which bind to Aβ and produce ROS and H_2_O_2_ with Aβ-plaque, lead to OS, affecting tau (T-proteins), and amyloid precursor proteins (APP),causing apolipoprotein-E (APOEAD) pathologies [40,51,52]. Tau-proteins contain three- or four-repeat microtubule binding-domains that are important in the formation of neuronal filaments [53]. A decrease in brain antioxidants and related enzyme competence in response to heavy metal-induced OS in zebrafish has been reported [54]. In progressive AD, the biometal homeostasis of ionic copper, iron, and zinc is disrupted by over-accumulation of neurotoxic insoluble Abeta fibrils generated from dityrosine cross-linked Aβ-peptides and proteins through a radical mechanism [36,54,55]. In the pathogenesis of AD, the coupling of tyrosine occurs under Cu (II)- and Fe (III)-induced OS [56]. This phenomenon generates H_2_O_2_, which is linked to neurotoxicity and neurodegeneration in the progression of AD [54].

In addition, other toxic metals (such as Pb, Sn, Hg, Mn, and Al) contribute to dementia in AD and cause OS, altering neural proteins and decreasing brain acetylcholine transferase level. Consequently, acetylcholine level declines in cholinergic neurons [40,51,52,56,57,58]. In AD, declines in acetylcholine level cause acetylcholinesterase activity to decline, leading to a decrease in the rate of acetylcholine hydrolysis into choline [51]. Therefore, in AD, acetylcholinesterase inhibitors are used to attenuate cognitive impairment [51]. An acetylcholinesterase inhibitor in red leaf tea extract, which is rich in delphinidin- and cyanidin-3-*O*-galactosides, has been reported [59]. The glucosides of ANTs are present in nutraceutical Medox, which is a berry extract with high ANT content. Medox is effective in alleviating mitochondrial dysfunction induced by Abeta toxicity and OS [60,61]. Increasing evidence shows that 3-*O*-β-glucopyranoside (Cy3G) from plants protects against Abeta 25–35 cytotoxicity, ROS, reactive nitrogen species (RNS), and neurodegenerative AD [61,62]. Purple rice extract, which is rich in cyanidin, also exerts a protective effect against Abeta 25–35 [31]. Thus, the nature of the glycoside may not be important in ANT activity. The mechanism of action of Cy3G involves the reticence binding of Abeta25–35 fibrils and the inhibition of ROS formation, leading to increased cognitive function [62].

OS is recognized as a major factor causing many pathologies, including AD. Toxic metals, including Cd, and toxic metal-induced OS have noxious effects on hypothalamic-pituitary-gonadal function, disrupting sex hormones. Elevated OS in neurological disorders generates ROS, and excessive RNS and cell signaling molecules lead to lipid, protein, and DNA peroxidation [63,64]. AD progression is complemented by an increase in the redox metals Fe and Cu, which can generate detrimental ROS. This process is completed by a decrease in cytochrome C oxidase, advanced glycation end-products, protein carbonyls bodies, 8-hydroxyguanidine, malondialdhyde (MDA), peroxynitrite, heme-oxygenase-1 (HO-1), and the formation of proinflammatory cytokines, such as interleukin-6 (IL-6). Increased levels of well-recognized neurogenic agents (including acrolein, 4-oxo-trans-2-nonenal (4-ONE), 4-hydroxy-trans-2-nonenal (HNE), and 4-oxo-trans-2-hexenal) are byproducts of augmented lipid peroxidation [65,66]. Tyrosine residues of aromatic amino acid are directly attacked by free radicals, leading to accumulation of dityrosine, 3-nitrotyrosine, and Abeta plaques in the brain with AD. This coupling is catalyzed by Cu (II) [55].

Single antioxidants and their combinations offer protection against AD. Cyanidin-3-*O*-glucoside is effective in alleviating the toxicity of Cd [67]. It might be possible to eliminate toxic metals through their chelation with ANTs [68]. In a study, commelinin, a blue pigment of *Commelina communis*, was shown to form a complex of four Mg ions and six ANT molecules [69]. Similarly, it has been reported that cyanosalvianin, a blue pigment from *Salvia uliginosa*, forms a complex of six molecules of ANT with two Mg ions [70]. Currently, two therapeutic metal chelators, coquinol and desferrioxamine, are being used to control AD, but with limited success [36]. Similarly, curcuminoids and ginkgo extracts have been examined, but with partial success [36]. Therefore, potent antioxidants and metal chelators, such as ANTs, may have a good potential for controlling AD. All these reports showed that, by forming a complex with toxic metals, ANTs could potentially be used to purge accumulated toxic metals in both the brain and body.

## 4. Anthocyanins and their Antioxidant Activity

The biological activities of ANTs are mainly exhibited through their antioxidant characteristics. The antioxidant activities of ANTs depend on several factors, including the position of the hydroxyl groups, glycosylation patterns, and the nature of the sugars [2]. Glycosylation of anthocyanidin affects its antioxidant capacity in various ways. First, by reducing the number of hydroxyl groups from which a radical can be displaced. Second, by reducing the metal chelation sites. Third, by altering the environment in which the oxidation occurs [71]. However, different patterns in glycosylation might increase or decrease the antioxidant capacity of ANTs and glycosides, and the antioxidant activities of aglycones might be similar [61,62,72,73,74]. However, glycosylation at the C3 of ANT contributes to its antioxidant capacity, with smaller number of glycosyl units at C3 enhancing the antioxidant capacity of ANT molecules [74,75]. Diglycosylation at C3 and C5 reduces the antioxidant capacity of ANT molecules [75].

Castaneda-Ovando et al. [3] suggested that free radicals are stabilized by the basic structure of anthocyanidin, and it produces quinonoid structures (Figure 4). Tsuda et al. [76] reported the mechanism of ANT-mediated radical-scavenging of the powerful oxidant peroxynitrite anion (O=NOO^−^). This action involves splitting ring B of ANT, as shown in Figure 5. Peroxynitrite is produced in vivo through interaction of superoxide and nitric oxide in every cell or tissue of the body to produce nitrotyrosine; misfolding of this protein leads to the formation of an AD plaque in the brain.

During Abeta plaque formation, the nitration/dimerization of tyrosine occurs in the order of cyanidin-3-rutinoside > malvidin-3-glucoside ≈ delphenidin-3-glucoside > petunidin-3-glucoside [75]. The glycosylation, hydroxylation pattern, and radical stabilization of ANTs directly affect their biological activity. Bioactive compounds from blueberry and blackcurrant exhibit different glycosylation patterns around ANT structures and possess strong antiproliferative and antioxidant capacities [64].

Various researchers have suggested that, after ingestion, the biological activity of ANTs is regulated by their metabolites [77,78]. Through non-antioxidant mechanisms, ANTs inhibit cyclooxygenase-2 (COX-2), inducible nitric oxide (iNOS) protein, and mRNA expression [79,80]. The inhibition of COX by cyanidin glycoside, which is present in berries and cherries, has also been reported [81]. Mulberries are a rich source of ANTs that induce antioxidant enzymes and promote cognition [82].

## 5. Anthocyanin Absorption, Bioavailability, and Metabolism

The absorption of ANTs and anthocyanidins depends on the dietary burden of fruits, vegetables, and wine. These molecules are primarily absorbed from the stomach or small intestine and are excreted via urine as glycosides or as glucuronides [43,83,84]. The absorption and bioavailability of ANTs from raspberries by patients with ileostomy, has been described by González-Barrio et al. [43]. After ingestion, the three ANTs found in raspberries (cyanidin-3-glucoside, cyanidin-3-sophoroside, and catechin), along with their metabolites, were not detected in the plasma of healthy or ileostomy volunteers; however, a low level of ANT (<0.1%) absorption in the small intestine was detected in healthy volunteers [85]. The very low absorption of ANTs might be sufficient to facilitate cell signaling, gene regulation, and other biological activities. This phenomenon might explain the health benefits of ANTs through a non-antioxidant mechanism [2]. ANT excretion via urine indicates little or no absorption [84]. However, in ileostomy volunteers, 40% of ANTs and 23% of ellagitannin sangullin H-6 were recovered in ileal fluid, indicating the hydrolysis of ellagitannins in the stomach and/or small intestine [84].

The degradation of ANTs to phenolic acid by colonic microbiota in the large intestine has also been reported [84,86,87]. Kawabata et al. [88] showed that intestinal microbiota are important in the metabolism of polyphenols and the chain fission products of pre-ANTs. Gut microbiota assist the breakdown of ANTs into protocatechuic acid and urolithins [89]. These studies showed that a complex interaction exists between dietary polyphenols and the host microbiome, and that the metabolism and microbial metabolites of ANT are present in the colon and systemic circulation. ANTs might also be involved in the activation of various metabolic enzymes, such as alpha-glucosidase, COX1, and COX2 [90,91]. Under physiological conditions, deglycosylation (aglycone) of ANTs produces unstable cyanidins, Figure 5 [92]. This phenomenon arises owing to the oxidation of phenolic hydroxyl groups to quinones, which have low biological activity [93]. Grape seed polyphenols (GSP) have protective therapeutic roles in AD and neurogenerative disorders. The interaction of GSP with intestinal microbiota converts dietary polyphenols into two phenolic acids, 3-hydroxybenzoic acid and 3-(3-hydroxyphenyl) propionic acid, in the brain. There is an evidence that both these phenolic acids interfere with the generation of the neurotoxic Abeta peptides involved in AD pathogenesis [38,94].

## 6. Alzheimer’s Disease and Anthocyanins

Multifactorial neurodegenerative AD has several overlapping pathways of expression. It progresses through the aggregation and deposition of Abeta1-42 and ROS-induced OS, causing excessive neuroinflammation [95]. Research on AD to develop therapies that inhibit filament formation of the tau-protein formed by OS in the brain should be a top-priority [53].

There are many therapeutic options to treat AD. Currently, there are five Food and Drug Administration (FDA)-approved medications to control AD. Three of these medications are donepezil, glutamine, and rivastigmine, all of which are cholinesterase inhibitors. The fourth drug is memantine, which exerts its effect through a different mechanism. The fifth drug is a combination of cholinesterase inhibitors (donepezil and memantine). For reversing AD, several natural products—including polyphenols such as curcumin, phytoestrogens, and antioxidants—have been studied.

A strong focus has been placed on natural polyphenolic ANT components. Many researchers have reported the inhibition of amyloid filament formation by berries that are rich in cyanidins and their glycoside, ANTs [53,95]. It is critical to discover suitable potent antioxidants that can protect astrocytes and brain tissue from OS. Date-palm fruit is a rich source of dietary fiber, as well as antioxidant ANTs and phenolic acids, including ferulic acid, protocatechuic acid, and caffeic acid. Following feeding with date fruits, rats that suffer from AD and severe anxiety behavior showed a decrease in Abeta, which lowered the risk of AD [96]. Many berries are rich in potent antioxidant polyphenolic ANTs that might offer protection from AD through different mechanisms [64,97,98]. For instance, ANTs in red raspberry and green tea are beneficial in reversing AD [99,100]. In another study, compared to transgenic AD mice fed blackcurrant, those fed bilberries showed a significant reduction in soluble Abeta 40 and Abeta 42 levels. In the cerebral cortex, blackcurrant and bilberry extract reduced APP levels in AD mouse models, but changes in the expression or phosphorylation of tau-protein were not observed [77].

ANTs cross the blood-brain barrier and protect brain tissue from Abeta toxicity, mitochondrial dysfunction, and apoptosis induced by OS [101]. A formulation of ANTs/anthocyanidins decreased tau-phosphorylation induced by Abeta (1–42) [100]. Isaak et al. [102] have reported that lingonberry ANTs (cyanidin-3-galactoside, cyanidin-3-glucoside, and cyanidin-3-arabinoside) protect cells from OS-induced apoptosis. Furthermore, Badshah et al. [103] showed that Abeta (1-42)-induced neurodegeneration in AD could be reversed by powerful antioxidants, including black soybean ANTs. AD might be reversed by ANTs through the mitochondrial apoptotic pathway, by regulating Bax, Cyto-C, caspases-9 3, tau-proteins, and BACE-1 [103,104]. Gutierres et al. [90] and Pacheco et al. [105] measured the levels of nitrite/nitrate (NOx), Na(+), K(+)-ATPase, Ca(2+)-ATPase, and acetylcholinesterase (AChE) activity in the cerebral cortex and hippocampus of AD in a mouse model. These authors showed that ANTs regulate the pumping activity of ions and cholinergic neurotransmission, reducing the onset of dementia. Protection of SH-SY5Y cells against amyloid Abeta (1–42) induces apoptosis by regulating Ca (2^+^) homeostasis, particularly in the presence of ANTs from *Aronia melanocarpa*. ANTs of this plant decrease intracellular calcium and ROS but increase ATP and mitochondrial potential. Gene transcription of ANTs present in this plant causes the upregulation of protein expression of calmodulin and Bcl-2; the down-regulation of cyt-C and caspase-9; and cleaves caspase-3 and Bax [106]. The ANT malvidin and its 3-glucoside (openin) show similar protective effects against Abeta (1–40)- and Abeta (25–35)-induced neurotoxicity, in addition to preserving Ca (2^+^) homeostasis and improving neuro-dysfunction [6].

Grapes belong to the berry family and are very rich in ANTs [99,107]. Grape seeds are rich in pro-ANTs, which protect against OS, lipid peroxidation, and DNA fragmentation. Grape seeds from 9 varieties of grapevine (*Vitis vinifera* L.) contain 22 different ANTs, with ANT levels ranging from 0.5 to 4.99 g kg^−1^. The type of ANT and its concentration depends on the variety and cultivation season. Peonidin, delphinidin, and malvidin are major ANTs in grapes [94,99].

The claim that berries promote mental health and protect against many diseases has led to a linear increase in the use of berries in health products [2,92,100]. The nutrients in berries might have other epigenetic effects on AD, contributing to slow down the progression of this disease [108]. Mitochondrial dysfunction and AD progression might be subdued by ANTs owing to their ability to inhibit apoptosis induced by Abeta, reduce ROS, and reduce intracellular calcium; however, ANTs also elevate ATP and mitochondrial membrane potential. This type of cytoprotective function in AD has been shown for ANTs found in *A. melanocarpa* (black chokeberry), which is very rich in antioxidants [106]. ANTs and other polyphenols are effective antioxidants that might inhibit AD and ROS [109].

## 7. Anthocyanin Nanoparticles

There are numerous reports of the therapeutic potential of ANT nanoparticles (An-NPs) in reducing AD pathology. An-NPs have the advantage of improving the efficiency of ANTs through nanodrug delivery systems [93]. An-NPs have better effectiveness, absorption, and bioavailability than parent ANTs [93]. Many different types of nanoparticles of ANTs have been prepared. Examples include biodegradable polymer-based polylactide-co-glycolide (PLGA), chitosan nanoparticles (CS-NPs), and polyethylene glycol (PEG)-encapsulated nanoparticles, which have good free radical-scavenging efficiency, high drug-loading efficiency, high stability, and potent water-soluble neuroprotective ANTs [93,110]. An-NPs are nontoxic to SH-SY5Y cells, with a favorable viability profile against Abeta and temper AD markers (APP). An-NPs also attenuate the protein expression of BACE-1 neuroinflammatory markers, such as phosphonuclear factor kB (p-NF-kB), tumor-necrosis factor (TNF-α), and inducible nitric oxide synthase (iNOS), as well as neurotic markers, such as Bax, Bcl2, and caspase-3,which are complemented in Abeta-induced neurodegeneration, in SH-SY5Y cells [93].

In 2019 [111], controlled drug delivery systems were developed by using nanoparticles that are considered neurotherapeutic. For instance, one report claimed that ANT nanoparticles control amyloid-beta (Abeta1-41) plaques and neurofibrillary tangles (NFTs), which are the main issues in AD. ANT PEG-gold nanoparticles (PEG-AuNPs) are considered effective in reducing neurological disorders in a mouse model of AD [111]. Mice injected with Abeta1-42 are protected by a treatment that contained ANT-loaded PEG-AuNP, which prevents apoptosis and neurodegeneration. The neuroprotective defense mechanism of PEG-AuNP might occur through protection of pre- and post-synaptic proteins from Abeta-induced synaptic dysfunction [112]. This phenomenon also prevents the hyperphosphorylation of tau-protein at serine 413 and 404 in the Abeta of AD mouse models [113]. In another report, the same authors showed that ANTs in Korean black beans have a neuroprotective capacity that regulates the phosphorylated-phosphoinositol 3-kinase-Akt-glycogen synthase kinase 3-beta (p-PI3/Akt/GSK3beta) pathways by stimulating the endogenous antioxidant system of nuclear factor erythroid 2-related factor-2 (Nrf2) and heme-oxygenase-1(Nrf2/HO-1). This phenomenon prevents apoptosis by suppressing the activation of caspase-3 and PARP-1 expression. ANTs in black bean also facilitate the expression of TUNEL and Fluoro-Jade B-positive neural cells in APP/PS1 animal models. The authors concluded that ANTs are potent antioxidant neuroprotective agents that diminish Abeta O-induced neurotoxicity in HT22 cells via PI3K/Akt/Nrf2 signaling [113].

The published literature highlighted in this review support the view that there are many ANTs and related polyphenolic natural products that may potentially be used as therapeutic approaches in AD research. These natural molecules could be further modified and exploited to enhance their distinct biological activities. The information accumulated in this review also supports that anthocyanins are protective molecules with good potential to alleviate neurodegenerative diseases, such as AD.

## 8. Conclusions

Most berries, fruits, and vegetables are exceptionally rich in anthocyanins, which are potent antioxidants. In AD, several Abeta polypeptides are oxidatively produced from APP. The medications currently used to combat AD partially inhibit the enzymatic and oxidative breakdown of acetylcholine and tau-proteins. Therefore, it is important to discover dominant antioxidants, such as anthocyanins, that are effective against neurodegenerative disorders and can help improve cognitive function. Anthocyanins fulfill this requirement, potentially representing an inexpensive way for treating aging-related neurodegenerative diseases, including AD. However, the metabolism rate of anthocyanins varies across individuals, which may affect their overall effectiveness.

## Figures and Tables

**Figure 1 molecules-24-04255-f001:**
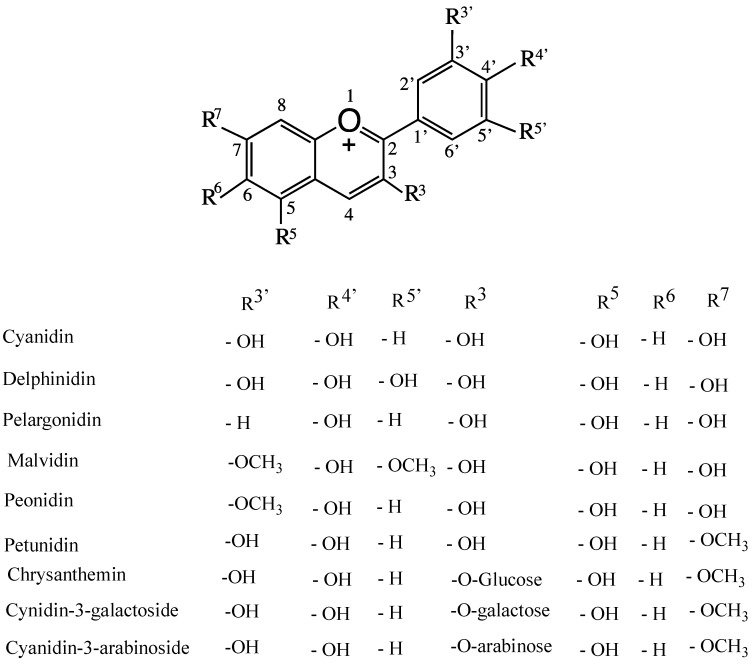
Naturally occurring substituted cyanidins and anthocyanidins.

**Figure 2 molecules-24-04255-f002:**
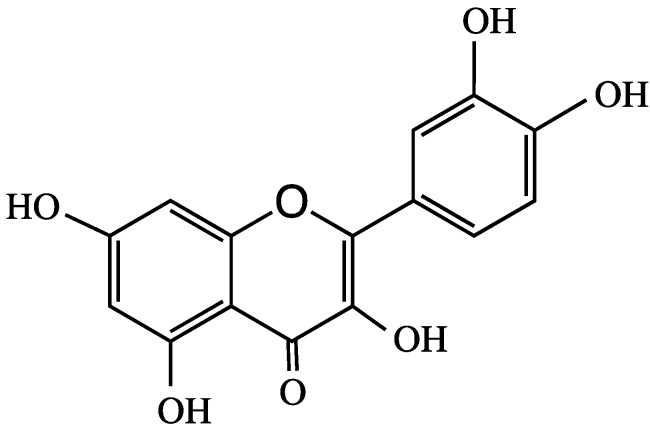
Molecular structure of quercetin.

**Figure 3 molecules-24-04255-f003:**
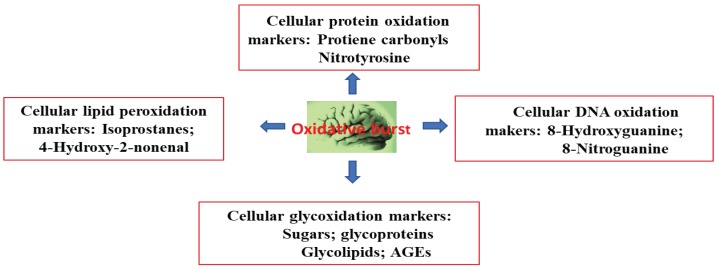
Oxidative burst produces different biomarkers in the brain.

**Figure 4 molecules-24-04255-f004:**
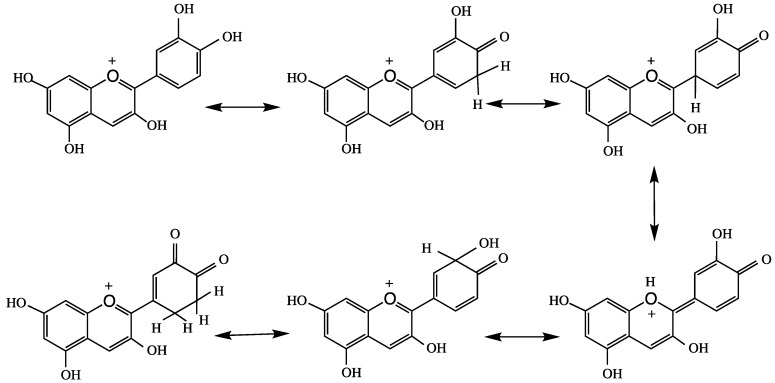
Quinonoid structures stabilize anthocyanidins.

**Figure 5 molecules-24-04255-f005:**
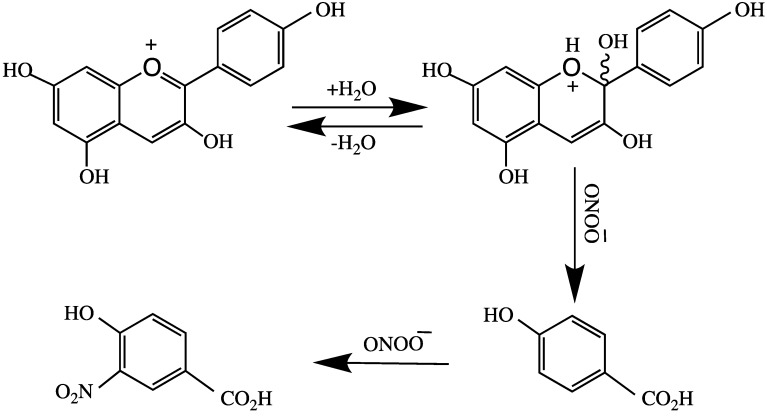
Peroxynitrite anion catalyzed splitting of ring B in anthocyanins.

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
