# Peer review of "Anthocyanins Potentially Contribute to Defense against Alzheimer’s Disease"

_molecules, 2019, doi:10.3390/molecules24234255_

Round 1

Reviewer 1 Report

The authors review the protective role of anthocyanins against Alzheimer's disease. However, the part devoted to the description of Alzheimer's disease mechanisms should be re-written and corrected. The review contains many incorrect or not precise information. Only some examples of the major points are:

- It is stated that no biomarckers exist for AD (line 127). However it is known that Abeta42 and tau levels in CSF predicts AD, though CSF sampling is not used systematically because it is an invasive method:

Hansson et al, CSF biomarkers of Alzheimer's disease concord with amyloid-β PET and predict clinical progression: A study of fully automated immunoassays in BioFINDER and ADNI cohorts, Alzheimers Dement. 2018 Nov;14(11):1470-1481.

- The authors write in lines 131-132 that "APP splits in fragments of Amyloid-beta peptide" which is "promoted by beta and gamma-secretase". The truth i that the cleavage of APP by beta and gamma-secretase is required to generate Abeta, it is not only promoted. Please change this pair of sentences. 

- Lines133: The authors state: “ This fragmentation causes the loss of APP, which is essential for neuronal growth, survival, and 134 repair [39].” It is NOT truth that in AD APP is lost. In dominant alzheimer’s disease cases, caused by mutation in APP and presenilin genes, there is an increased of the amyloidogenic processing of APP and a decreased in the non-amyloidogenic processing of APP (alpha-secretase). But the total levels of APP do not change or even increase.

- The review states in line 146 that "Astrocytes are also involved in the expression of apolipoprotein E and the degradation and removal of Abeta1-42, indicating that astroglia occurs during the progression of AD”. What does it mean astroglia occurs? It should say astrogliosis?"

Author Response

The authors review the protective role of anthocyanins against Alzheimer's disease. However, the part devoted to the description of Alzheimer's disease mechanisms should be re-written and corrected. The review contains many incorrect or not precise information. Only some examples of the major points are:

- It is stated that no biomarckers exist for AD (line 127). However it is known that Abeta42 and tau levels in CSF predicts AD, though CSF sampling is not used systematically because it is an invasive method:

Hansson et al, CSF biomarkers of Alzheimer's disease concord with amyloid-β PET and predict clinical progression: A study of fully automated immunoassays in BioFINDER and ADNI cohorts, Alzheimers Dement. 2018 Nov;14(11):1470-1481.

Response: The use pf PET scan to measure  tau/Abeta ratio and histological brain scans for the diagnosis of AD have been introduced in the text and citation of Hansson et al. (2008) has been added (Line 145-147).

The authors write in lines 131-132 that "APP splits in fragments of Amyloid-beta peptide" which is "promoted by beta and gamma-secretase". The truth i that the cleavage of APP by beta and gamma-secretase is required to generate Abeta, it is not only promoted. Please change this pair of sentences.

Response: A sentence "The cleavage of transaminase protein, amyloid precursor protein (APP), is augmented by beta- and gamma-secretase to generate ABeta, with 39-43 amino acids" has been added (line 149-151).

Lines133: The authors state: “ This fragmentation causes the loss of APP, which is essential for neuronal growth, survival, and 134 repair [39].” It is NOT truth that in AD APP is lost. In dominant alzheimer’s disease cases, caused by mutation in APP and presenilin genes, there is an increased of the amyloidogenic processing of APP and a decreased in the non-amyloidogenic processing of APP (alpha-secretase). But the total levels of APP do not change or even increase.

Response: A sentence "The amyloidogenic processing of the APP increases and the non-amyloidogenic processing of APP (alpha-secretase) decreases. However, the total levels of APP do not change or even increase" has been added (Line 152-154).  

The review states in line 146 that "Astrocytes are also involved in the expression of apolipoprotein E and the degradation and removal of Abeta1-42, indicating that astroglia occurs during the progression of AD”. What does it mean astroglia occurs? It should say astrogliosis?"

Response: The word astroglia has been replaced by astrogliosis (Line 170-171)

Reviewer 2 Report

This is a very interesting and complete review. the authors must be revised the references. Year must be in bold.

Author Response

The publication year has been made bold in the citation list.

Reviewer 3 Report

Review for molecules-614277 and manuscript entitled: “Anthocyanins potentially contribute to defense against Alzheimer’s Disease” In my opinion the manuscript can be accepted for publication after improvements that is below.

Detailed suggestion:

The contents for the review article conceived are very well, and the paragraphs are described as well organized. However, the descriptions of the relationship between Alzheimer's disease and oxidative stress would be better if there were graphics and tables assistance in Section 3.

Author Response

The impact of oxidative burst on brain cellular oxidative markers is shown in Fig. 3

Reviewer 4 Report

Although the matter of the review is widely treated and discussed in literature, the article is interesting and pleasant to read. Only few comments: in my opinion, the Authors reveal too much certainties about the origin and causes of the AD. In some cases, as can be read in literature, causes and effects of the pathology are interchangeable. Thus, I suggest to smooth these statements.

Some English typos should be modified.  

Author Response

The interchangeable causes and effects has been elaborated and the manuscript language has been extensively edited by professional editors.

Round 2

Reviewer 1 Report

Please, change the sentence  "The cleavage of transmembrane protein, amyloid precursor protein (APP), is augmented by beta- and gamma-secretase to generate ABeta, with 39-43 amino acids" (line 149-151) for the following sentence: "The transmembrane protein amyloid precursor protein (APP) is cleaved by beta- and gamma-secretase activities to generate Abeta, with 39-43 amino acids" Please, change the sentence "The amyloidogenic processing of the APP increases and the non-amyloidogenic processing of APP (alpha-secretase) decreases. However, the total levels of APP do not change or even increase" for the following sentence: "The amyloidogenic processing of the APP towards the more toxic Abeta42 is increased in dominant forms of AD, while no changes in total APP levels occur"

Author Response

The document has been checked for minor spelling errors and have been corrected. The corrections are highlighted yellow. The sentences line 139-143 have been replaced with the suggested sentences (line 145-148; highlighted green).  Fig. 3 has been replaced.

        Thank you for your helpful suggestions and comments.

This manuscript is a resubmission of an earlier submission. The following is a list of the peer review reports and author responses from that submission.